# Combined Peptidomics and Metabolomics Analyses to Characterize the Digestion Properties and Activity of *Stropharia rugosoannulata* Protein–Peptide-Based Materials

**DOI:** 10.3390/foods13162546

**Published:** 2024-08-15

**Authors:** Wen Li, Wanchao Chen, Zhong Zhang, Di Wu, Peng Liu, Zhengpeng Li, Yan Yang

**Affiliations:** Institute of Edible Fungi, Shanghai Academy of Agricultural Sciences, Shanghai 201403, China; liwen3848@126.com (W.L.); chenwanchao@saas.sh.cn (W.C.); zhangz0815@126.com (Z.Z.); vivian218074@163.com (D.W.); lipeng0504@126.com (P.L.); lizp_ln@126.com (Z.L.)

**Keywords:** protein peptides, simulated digestion, ACE inhibitory activity, metabolite markers, regulation pathways

## Abstract

Protein–peptide-based materials typically possess high nutritional value and various physiological regulatory activities. This study evaluated the digestion, metabolism, and activity of *Stropharia rugosoannulata* protein–peptide-based materials. After the *S. rugosoannulata* protein–peptide-based materials were digested (simulated) orally, in the stomach, and in the intestines, the proportions of >10,000 Da, 5000~10,000 Da, and <180 Da in the digestion products increased, and the peptide content was maintained at more than 120 mg/g dry weight. The digestion products of eight test groups with different oral–gastrointestinal digestion-level settings all had suitable ACE inhibitory activity (IC_50_ range 0.004~0.096 mg/mL). The main metabolite groups were lipid-like molecules, fatty acids, carboxylic acids, their derivatives, amino acids, peptides, and analogs. Bile and glycosylated amino acids were the main compounds that caused differences between groups. KEGG pathways enriched in differentially expressed metabolites included eight significantly upregulated pathways, including valine, leucine, and isoleucine biosynthesis, etc., and six significantly downregulated pathways, including the citric acid cycle (tricarboxylic acid cycle), etc. The arginine and proline metabolism pathways and the aminoacyl-tRNA biosynthesis pathways were upregulation and downregulation pathways that enriched multiple differentially expressed metabolites. Twenty-six metabolites, including bile acids, total bile acids, and the essential amino acids L-isoleucine and L-leucine, were differentially expressed metabolite markers of the protein–peptide-based material oral–gastrointestinal digestion products.

## 1. Introduction

Protein–peptide-based materials are a mixture of a high protein and peptide contents, and they show great potential for food, nutraceuticals, and pharmaceuticals. Protein–peptide-based materials usually have a good nutritional value and perform a variety of physiological regulatory activities. Protein peptides perform nutritional activities that provide the body with nutrients, such as proteins, peptides, and amino acids, to maintain and safeguard the health of the body. Protein peptides perform biological activities (Appendix A), such as antioxidant [1,2], anti-cell damage [3], anti-tumor [4], anti-hyperuricemia [5], anti-osteoporosis [6], anti-obesity [7], anti-bacterial [8,9,10], anti-inflammatory [11], anti-exercise fatigue [12], anti-aging [13], improvements in cognitive disorders [14], improvements in learning and memory [15], improvements in mitochondrial function [16], improvements in malnutrition [17], protection of the liver and kidneys [18,19], regulation of the intestinal flora [20,21], regulation of immunity [20], lowering of blood pressure [22], lowering of blood lipids [23,24,25], and lowering of blood glucose [21,26]. This means that they can be used as a functional ingredient and auxiliary drug to achieve disease prevention and treatment. With the growing concern for nutritional and functional health diets, the development of food-derived protein peptide products has become an increasingly important focus of food science research.

*Stropharia rugosoannulata* is a nutritious and tasty edible mushroom. The fruit body of *S. rugosoannulata* is rich in protein (about 50% dry weight) and peptides (about 14% dry weight). The proteins of *S. rugosoannulata* are high-quality proteins rich in many essential amino acids. Its rich methionine, cysteine, and lysine content can compensate for plant protein. The peptides of *S. rugosoannulat* perform better biological activities, such as lowering blood pressure, lowering blood lipids, lowering blood glucose, and providing antioxidants. Our previous study found that the ultrasound-assisted enzymatic hydrolysis of *S. rugosoannulata* protein–peptide-based materials had a high yield, which could reach 49~52%. The total amount of protein and peptides in the base materials was more than 60%, and the peptide content was 492.87 mg/g of dry weight [27], while the protein content of the unhydrolyzed peptide was 172.19 mg/g of dry weight. The protein–peptide-based materials had superior angiotensin-converting enzyme (ACE) inhibition and blood-pressure-lowering activity, with the ACE half-maximal inhibitory concentration (IC_50_) in a range of 0.071–0.123 mg/mL [28,29,30], and its blood-pressure-lowering activity was comparable to that of the drug Benazepril hydrochloride. Protein–peptide-based materials have a pleasant taste. At a 1 mg/mL concentration, their salinity is equivalent to the salinity of the aqueous sodium chloride solution of the same concentration by 1.9~2.3-times [31]. The protein–peptide-based materials of *S. rugosoannulata* have a good application value in the nutritional supply of proteins and peptides, assisting antihypertensive therapy and sodium salt substitution.

The oral gastrointestinal tract digests and absorbs protein–peptide-based materials to perform nutritional and functional activities. Mao et al. [32] explored the purple-speckled kidney bean protein’s antioxidant activity and structural characteristics during simulated gastric digestion *in vitro*. The results showed that peptides with low molecular masses were produced, resulting in increased antioxidant activity in the digestion product. At the end of digestion, the total antioxidant capacity and ferric ion-reducing power increased by 296.97% and 54.01%, respectively, compared to protein. The digested protein exhibited granules arranged in a disorderly manner to form a dense reticular structure under scanning electron microscopy (SEM). Shi et al. [33] studied the relationship between rice glutelin structure and its hydrolysate’s antioxidant and ACE inhibitory activity at different *in vitro* simulated digestion times under different hydrolysis times. The ACE inhibitory activity increased, reaching the maximum at a hydrolysis time of 2 h, with an IC_50_ of (0.693 ± 0.011) mg/mL, then decreased. The antioxidant capacity of the hydrolysate from a hydrolysis time range of 0.5 to 2 h was negatively correlated with the ACE inhibitory activity, and both decreased. Jiao et al. [34] analyzed quinoa albumin’s antioxidant activity and structural changes during *in vitro* simulated digestion. After simulated gastrointestinal digestion, the DPPH· and ABTS^+^· scavenging rates, total reducing power, Fe^2+^ chelation rate, and Cu^2+^ chelation rate of albumin digestion products were 34. 85%, 68. 62%, 0. 179, 54. 52%, and 7. 76% higher, respectively, than those of undigested albumin. SEM observation showed that, after pepsin hydrolysis, the digested particles became significantly smaller, disorderly, and arranged in a dense structure. The surface area and the pore size of the digested particles increased. It was found that, after the *in vitro* gastrointestinal-simulated digestion of the rice protein hydrolysate and milk protein hydrolysate, the number of peptide molecules with a molecular weight less than 200–500 Da and the ACE inhibitory activity increased in the digested products [35,36]. The application scenarios of the protein–peptide-based materials of *S. rugosoannulata* include the fields of health food, flavored food, and food additives. As a healthy food, the composition changes in protein–peptide-based materials after oral gastrointestinal digestion are still unknown, and whether the digestion products can still exert ACE inhibitory activity and nutritional effects (peptide and amino acid supplements) on the organism is unclear. Therefore, analyzing the digestion properties of protein–peptide-based materials is crucial for applying base materials as a healthy food. It is significant to explore the digestion and metabolism of protein–peptide-based materials and clarify the scientific basis for their nutritional supply and functional activity, which is essential for developing food-derived protein–peptide-based foods.

To understand the changes in the metabolites produced during the oral–gastrointestinal digestion of the protein–peptide-based material, a combined targeted and nontargeted metabolomic analysis was used for the comprehensive analysis of the digestion products. Targeted metabolomics (peptidomics) mainly focused on the analysis of peptide molecules, specifying the critical metabolite peptide molecules of the protein–peptide-based material. Nontargeted metabolomics was used to analyze the dynamic changes in the endogenous small-molecule compounds with a relative molecular weight of ≤1000 Da and screen for differentially expressed metabolites through confidence analysis. Through the nontargeted metabolomics analysis, we were able to identify the pathways of the differentially expressed metabolites to reveal the metabolic markers.

We also conducted an *in vitro* digestion simulation of *S. rugosoannulata* protein–peptide-based materials using an adult digestion electrolyte solution. Targeted peptidomics analysis was used to obtain information about the distribution of peptide molecules, and nontargeted metabolomics was used to analyze the nutrient metabolite and metabolic pathways generated in the digestion products. The morphology, molecular weight distribution, and ACE inhibitory activity of the obtained digestion products were also analyzed. Information about the metabolism and activity of the protein–peptide digestion products obtained in this study can provide a theoretical basis for the development and application of *S. rugosoannulata*-based protein–peptide products.

## 2. Materials and Methods

### 2.1. Materials

*S. rugosoannulata* protein–peptide-based materials were prepared as described in our previous study [30]. Mushroom (mushroom strain NCBI No. SRR14469700), at a concentration of 48 g/L, was added to a solution of 2 × 10^5^ U/g alkaline protease (1% *w*/*w*) (Beijing Solarbio Science & Technology Co., Ltd., Beijing, China), and hydrolysis was conducted at a hydrolysis temperature of 42 °C and a pH of 8.5 under synchronous ultrasonic-assisted enzyme hydrolysis conditions (bath ultrasonic power density 120 W/L, ultrasonic frequency 20 kHz) for 40 min to obtain an enzymatic hydrolysis solution of *S. rugosoannulata*. The hydrolysis supernatant was collected by centrifuging the hydrolysis solution at 9000× *g* for 10 min at 4 °C and ultrafiltration through a GE 3000 NMWC UF membrane. The permeate was freeze-dried at −70 °C for 48 h to obtain the protein–peptide-based materials.

Adult digestion electrolyte solutions, saliva digestion simulated solution (containing α-amylase, 150 U/mL; pH 7.0 ± 0.1), gastric digestion simulated solution (containing pepsin, 4000 U/mL; pH 1.6 ± 0.1), and intestinal digestion simulated solution (containing trypsin, 200 U/mL; bile salt, 20 mM; pH 7.0 ± 0.1) were purchased from Xiao Dong Pro-health Instrumentation Co Ltd. (Suzhou, China). The concentrations of electrolytes in the simulated digestion fluids are shown in Appendix A. The standards of cytochrome C (molecular weight 12,384), peptidase (molecular weight 6500), bacillus peptide (molecular weight 1422), ethionine–ethionine–tyrosine–arginine (molecular weight 451), and ethionine–ethionine–ethionine (molecular weight 189) were purchased from Shanghai Yuanye Bio-Technology Co., Ltd. (Shanghai, China).

### 2.2. In Vitro Simulated Digestion of Protein–Peptide-Based Materials

Food digestion after oral intake includes both gastric and intestinal digestion. The digestion time is around 2–4 h in a healthy digestive system. The reactive material–liquid ratio of the protein–peptide-based materials for *in vitro* simulated digestion was determined according to the recommended daily food intake and the typical digestive juices secreted from the human oral cavity, stomach, and intestines.

Eight simulation digestion test groups (groups 1–8) with different conditions for gastrointestinal digestion were set up. Thus, 5 g of the protein–peptide-based materials was placed into a 250 mL triangular flask, and 10 mL of saliva digestion simulated solution was added. The mixture was left to react for 2 min at 37 °C in a thermostatic shaker (120 rpm). Next, 100 mL of gastric digestion simulated solution was added, and the mixtures for test groups 1–8 were reacted for 30 min, 60 min, 90 min, 120 min, 150 min, 180 min, 210 min, and 240 min, respectively. After the above reaction, 100 mL of intestinal digestion simulated solution was added. The reaction times after the addition of intestinal digestion simulated solution to the reaction solution were 210 min, 180 min, 150 min, 120 min, 90 min, 60 min, 30 min, and 2 min for test groups 1–8, respectively. At the end of the reaction, the solution was lyophilized at −70°C for 24 h to obtain the digestion products. The digestion products from groups 1 to 8 were numbered 1–8 for targeted peptidomics analysis and A–H for nontargeted metabolomics analysis; undigested samples were labeled “CK”.

### 2.3. Morphological Analysis of Digestion Products

The morphology of the digestion products was observed via SEM. The product powder was immobilized on a conductive adhesive with a voltage of 5.0 kV, a lens height of 9.8 mm, and magnifications of 200×, 500×, 1000×, and 2000×.

### 2.4. Molecular Weight Distribution Analysis of Digestion Products

SDS-PAGE gel electrophoresis and HPLC analysis were performed to determine the molecular weight distribution of the digestion products. In brief, 30 mg of digestion products was placed into a 1.5 mL polypropylene centrifuge tube, and 500 μL of pre-cooled SDS lysis solution was added. The mixture was then vortexed until the products had dissolved, followed by centrifugation at 12,000× *g* and 4 °C for 15 min. Next, 20 μL of the sample solution was transferred to a 200 μL PCR tube, and 4 μL of protein loading buffer (Beyotime 6×, Shanghai SenGene Biotech Co., Ltd., Shanghai, China) was added. The sample solution was then vortexed, followed by centrifugation at 12,000× *g* and 4 °C for 5 min to collect the supernatant. The PCR tube containing supernatant was heated at 99 °C for 8 min, then centrifuged at 12,000× *g* and 4 °C for 5 min. The separator and concentrator gels for SDS-PAGE were prepared as shown in Appendix A. After preparation of the gels, 15 μL of the sample was added to the gel lane, followed by electrophoresis at a voltage of 100 V for the concentrator gel and 120 V for the separator gel. After electrophoresis, the gel was stained with Caulmers Brilliant Blue for 180 min; then, Caulmers Brilliant Blue decolorizing solution was added, and the gel was left to stand overnight on a shaking table. The gel was then washed with water and photographed.

The molecular weight distribution of the digestion products was analyzed using a Waters 2695 HPLC system equipped with a TSKgel G2000 SWXL column (300 mm × 7.8 mm, 5 μm). The sample was transferred to a 10 mL volumetric flask containing 100 mg of digestion products, and the mobile phase was diluted to scale. After centrifugation at 3000× *g* for 5 min and filtration through a 0.22 μm microporous membrane, the sample was injected into the HPLC system. The mobile phase was acetonitrile/water/trifluoroacetic acid (40/60/0.1 *v*/*v*), and the HPLC conditions were as follows: flow rate, 0.5 mL/min; UV detection wavelength, 220 nm; column temperature, 30 °C; sample injection volume, 10 μL. Digestion product molecular-weight-corrected standard curves were plotted using cytochrome C, peptidase, bacillus peptide, ethionine–ethionine–tyrosine–arginine, and ethionine–ethionine–ethionine standards.

### 2.5. ACE Inhibitory Activity Analysis of Digestion Products

The kit method was used to conduct the analysis of ACE inhibitory activity of the digestion products according to a procedure described previously [30]. In brief, 0.1 g of digestion products was dissolved in 50 mL of pure water. Then, the solution was serially diluted to obtain solutions with concentration gradients of 2.0, 0.4, 0.08, 0.016, and 0.0032 mg/mL. The ACE inhibitory activity of the solutions was measured using a DOJINDO ACE Kit-WST kit (Shanghai Youlu Biotechnology Co., Ltd., Shanghai, China). The ACE inhibitory activity was calculated by detecting the absorbance at 450 nm of 3-hydroxybutyric acid, which is produced by ACE-catalyzed 3-hydroxybutyryl-Gly-Gly-Gly. The ACE inhibitory activity curve was constructed using the sample concentration and inhibition rate. The *in vitro* ACE inhibitory activity experiment was conducted in triplicate, and the 50% inhibition rate was calculated.

### 2.6. Targeted Peptidomics Analysis of Digestion Products

Peptide content in the digestion products was analyzed according to the kit method (Suzhou Michy Biomedical Technology Co., Ltd., Suzhou, China) using a procedure described previously [37]. In brief, 0.1 g of digestion sample was added to 1 mL of kit extract (10% TCA) and homogenized on ice for 30 min. The sample was then centrifuged at 12,000× *g* and 4 °C for 10 min, and the supernatant was collected. Then, 10 μL of the supernatant was added to 190 μL of kit working solution and reacted at 60 °C for 30 min in a water bath. The absorbance at 562 nm was then measured. The kit working solution was a 50:1 mixture of kit reagents A and B. Reagent A was a bicinchoninic acid, Na_2_CO_3_, C_4_H_4_Na_2_O_6_, NaOH, NaHCO_3_ solution (pH 11), and reagent B was a CuSO_4_·5H_2_O solution.

Targeted peptidomics was analyzed using a method described previously [28]. A Ziptip C_18_ column was used for the desalting pretreatment of the digestion products. The samples were eluted with 60% ACN/0.1% TFA and then dried in vacuum to obtain the pretreated samples. Liquid chromatography–tandem mass spectrometry (LC–MS/MS) (Thermo Easy NLC liquid chromatograph, Thermo Scientific Q-Exactive Orbitrap mass spectrometer, 75 µm I.D. × 150 mm Acclaim PepMap RSLC C_18_ Nanoviper chromatographic column, Thermo Fisher Scientific, Roskilde, Denmark) was used to analyze the peptide sequences of the pretreated samples. The pretreated samples were dissolved in 20 μL of 0.1% formic acid and 5% acetonitrile, oscillated in a vortex, and then centrifuged at 8000× *g* and 4 °C for 20 min. The supernatant was collected, and 8 μL of the supernatant was injected for mass spectrometry identification. For liquid chromatography, 0.1% formic acid was used as mobile phase A, and 0.1% formic acid and 80% acetonitrile were used as mobile phase B. The LC parameters were 0 min, 97% A; 3 min, 97% A; 7 min, 92% A; 46 min, 68% A; 51 min, 56% A; 56 min, 99% B; 60 min, 99% B; 60.1 min, 97% A; and 70 min, 97% A. The flow rate of the mobile phase was 400 nL/min. The MS analysis conditions were as follows: primary mass spectral resolution, 120,000; automatic gain control, 4 × 10^5^; ion maximum injection time, 50 ms; mass scan range, 350–1550 *m*/*z*; secondary mass resolution, 30,000; automatic gain control, 1 × 10^5^; maximum ion injection time, 100 ms; number of ions selected for secondary fragmentation in primary mass spectra, 20; and NCE mode collision energy, 32.

### 2.7. Nontargeted Metabolomics Analysis of Digestion Products

Nontargeted metabolomics was used to characterize other small molecules in the digestion products. Eight digestion product samples (6 parallel samples in each simulation digestion test group: 54 samples) were pretreated and metabolite extracted as described above. Sample pretreatment was performed as follows: 60 mg of sample was added to a 1.5 mL centrifuge tube along with two tiny steel beads and 600 μL of extraction solution (methanol/water, *v*/*v* 7:3, containing L-2-chlorophenylalanine, 4 μg/mL). The samples were then pre-cooled in a refrigerator at −40 °C for 2 min, milled in a grinder for 2 min (60 Hz), ultrasound-extracted in an iced water bath for 30 min, and finally rested overnight at −40 °C. After centrifuging at 13,000× *g* for 10 min at 4 °C, 150 μL of the supernatant was filtered through a 0.22 μm organic-phase pinhole filter, and the filtrate was transferred to an LC injection bottle and stored at −70 °C. The quality control samples were made by mixing all sample extracts in equal volumes. The analyzing instrument was an ACQUITY UPLC I-Class coupled with an ACQUITY UPLC HSS T3 column (100 mm × 2.1 mm, 1.8 μm) and a QE high-resolution mass spectrometer (Thermo Fisher Scientific, Shanghai, China). The UPLC analysis conditions were as follows: column temperature, 45 °C; binary gradient elution mobile phases, water (phase A, containing 0.1% formic acid) and acetonitrile (phase B); flow rate, 0.35 mL/min; injection volume, 2 μL; elution programs, 0 min, 5% B; 2 min, 5% B; 4 min, 30% B; 8 min, 50% B; 10 min, 80% B; 14 min, 100% B; 15 min, 100% B; 15.1 min, 5% B; and 18 min, 5% B.

The MS analysis conditions were as follows: the sample mass spectrometry signals were collected in positive and negative ion scanning modes using an ESI ion source, a mass spectrometry scanning range of 100–1200 *m*/*z*, a full scanning resolution of 70,000, a secondary mass spectrometry scanning resolution of 17,500, spray voltages of 3800 V (in positive ion mode) and −3000 V (in negative ion mode), a sheath gas flow rate of 35 arb, an auxiliary gas flow rate of 8 arb, and a capillary temperature of 320 °C.

### 2.8. Statistical Analysis of the MS Data

The peptide sequences obtained from the targeted peptidomics were analyzed using Peaks 10.5 software and mapped to the UniProtKB precursor protein sequence database. Twenty-six protein sequences were obtained by searching the keywords “*Stropharia rugosoannulata*” (https://www.uniprot.org/, accessed on 30 June 2023).

The nontargeted metabolomics data processing software Progenesis QI v.3.0 was used for baseline filtering, identification, integration, retention time correction, peak alignment, and normalization of the acquired LC–MS raw mass spectral information. The main parameters were set as follows: precursor ion tolerance, 5 ppm; fragment ion tolerance, 10 ppm. The compound identification was based on the exact mass number, secondary fragmentation, and isotopic distribution. The HMDB (https://hmdb.ca, accessed on 29 December 2023), Lipidmaps (http://www.lipidmaps.org/, accessed on 29 December 2023), and Metlin (https://metlin.scripps.edu/, accessed on 29 December 2023) databases were used for compound characterization.

## 3. Results

### 3.1. Morphological Analysis of Digestion Products

The SEM morphology observation of the digestion products, obtained from the 4 h *in vitro* oral–gastrointestinal digestion simulation of the *S. rugosoannulata* protein–peptide-based materials, showed that the digestion products had a loose, rough, porous structure and showed apparent granular morphology with uneven particle sizes and dense distribution. Internal cavities or laminar structures were observed in some particles (Figure 1).

### 3.2. Molecular Weight Distribution Analysis of Digestion Products

The SDS-PAGE gel electrophoresis results for the different test sample groups are shown in Figure 2. The molecular weight distribution from the HPLC analysis is shown in Figure 3. No significant differences in molecular weight were detected between the digestion product sample groups, but there was a significant difference in molecular weight between the digestion product sample group and the undigested control sample group. According to the peak appearance of different molecular weight segments, the molecular weight range of the digestion products was divided into eight intervals: >10,000, 5000–10,000, 3000–5000, 2000–3000, 1000–2000, 500–1000, 180–500, and <180 Da. The molecular weight distribution of the digestion products according to these eight interval classifications is shown in Figure 4. The results of peak area integration revealed that after the protein–peptide-based materials were digested by oral–gastrointestinal simulation, the percentage of components distributed in the >10,000 (molecular weight did not exceed 18,000 Da), 5000–10,000, and <180 Da groups was greater than that in the undigested control samples, whereas the proportion of other molecular weights of components (3000–5000, 2000–3000, 1000–2000, 500–1000, and 180–500 Da) was lower than that of the undigested control samples. Appendix A shows the “Weight Average Molecular Weight (Mw)” and “Number Average Molecular Weight (Mn)” of the products’ different molecular weight segments. We considered that the change in the percentage of the >5000 Da fraction may be related to the production of this molecular weight fraction after the digestion of unhydrolyzed proteins by oral–gastrointestinal proteases. In comparison, the <5000 Da fraction in the base materials produced lower molecular weight peptides and amino acids by protease hydrolyzation, which led to a decrease in the proportion of the fractions in this molecular weight category and an increase in the proportion of fractions of <180 Da.

### 3.3. ACE Inhibitory Activity Analysis of Digestion Products

The peptide content and ACE inhibitory activity of the digestion products obtained from the *in vitro* oral–gastrointestinal simulated digestion of the protein–peptide-based materials were analyzed, and the results are shown in Figure 5. The results showed differences in the peptide content of the digestion products obtained at different levels of gastrointestinal digestion. However, the peptide content of the digestion products was maintained above 120 mg/g (dry weight). The digestion products had good ACE inhibitory activity at mass concentrations of 0.4 mg/mL, 2.0 mg/mL, and 10 mg/mL, with the IC_50_ ranging from 0.004 mg/mL to 0.096 mg/mL. The ACE inhibition rate reached >60% at 0.4 mg/mL. We hypothesized that the protein–peptide molecules in the base materials could still be converted into peptide molecules with good ACE inhibitory activity after successive oral–gastrointestinal simulation digestion.

### 3.4. Targeted Peptidomics Analysis of Digestion Products

The distribution of peptide spectra in the digestion products sample group is shown in Figure 6. The results showed that the mass spectral peak areas of the peptide molecules in the products were highly variable under different levels of oral–gastrointestinal digestion. The number of peptide molecules and the peptide spectral abundance of the products dominated by gastric digestion (gastric digestion time of more than 2 h, sample groups 5–8) were generally higher than those of the products dominated by intestinal digestion (intestinal digestion time of more than 2 h, sample groups 1–3), indicating that the base materials produced more abundant peptide molecules during the gastric digestion stage. In addition, the chain length analysis of the peptide molecules showed that long-chain peptides (tetradecapeptides and above) accounted for >10% of the gastric digestion-dominated products, indicating that gastric digestion had limited degradation ability for molecules of this molecular weight. The molecular weight was reduced in the intestinal digestion-dominated products, which were transformed into short-chain peptide molecules. The number of shared peptide molecules in the digestion products obtained from different gastrointestinal digestion levels was 32 (the number of peptide molecules obtained at different digestion levels was in the hundreds), and information on the shared peptide molecules is shown in Appendix A. Hexapeptide–octapeptide molecules accounted for 66% of the common peptide molecules.

The activities of shared peptide molecules were predicted using the BIOPEP-UMW database (https://biochemia.uwm.edu.pl/biopep-uwm/, accessed on 16 February 2024), and the results showed that the peptide molecules were potential ACE inhibitors. Even the peptide sequences that were digested by gastrointestinal proteases (virtually digested by pepsin, EC 3.4.23.1; trypsin, EC 3.4.21.4 in BIOPEP-UMW web) released diverse and abundant molecular fragments with ACE inhibitory activity, indicating that both the peptide molecules and their degradation fragments possessed good ACE inhibitory activity. A total of 161 fragments (94 species) showed ACE inhibitory activity in the peptide molecules. The peptide molecules were resistant to protease digestion in the *in vitro* oral–gastrointestinal simulated digestion assay, and it is expected that the peptide molecules with ACE inhibitory activity would contribute to the digestion products’ *in vivo* ACE inhibitory effect.

### 3.5. Nontargeted Metabolomics Analysis of Digestion Products

Mass spectrometry stability assessment was performed based on the distribution of ion intensity values for the nontargeted metabolomics analysis of the digestion products. An ion intensity box plot from the mass spectrometry stability assessment is shown in Appendix A. The results showed that samples in the same group were closely clustered, and the sample detection stability was good.

After processing raw data using metabolomics processing software and database comparison for compound characterization, ion peaks with missing values of >50% in the sample group were deleted, and compounds with secondary fragmentation characterization scores of 36 or more were screened. A total of 11,677 metabolites were extracted in positive and negative ion scanning modes, of which 6367 compounds were extracted in the positive ion scanning mode and 5310 compounds were extracted in the negative ion scanning mode.

The extracted metabolites were categorized, and the results are shown in Figure 7. The results showed that the primary metabolites in the superclass were lipids and lipid-like molecules (33.14%); the metabolites with the highest proportion in the secondary class were mainly fatty acyls (13.99%) and carboxylic acids and their derivatives (9.06%), and the primary metabolites in the subclass were amino acids, peptides, and their analogs (8.26%).

Principal component analysis (PCA) was used to observe the overall distribution of each sample group, and the distribution of samples in the PCA quadrant spaces is shown in Figure 8. The results showed that the undigested base materials (CK), product samples dominated by gastric digestion (samples E–H), and product samples dominated by intestinal digestion (sample A–C) were distributed in different quadrant spaces. Therefore, the samples that had received different levels of digestion could be effectively distinguished.

Orthogonal partial least squares discriminant analysis (OPLS-DA) was used to compare the differences in metabolic profiles between the sample groups, and the OPLS Da score plots are shown in Appendix A. The results showed that the sample comparison groups were significantly different within 95% confidence intervals (in the case of groups vs. CK). The OPLS Da model parameters are shown in Appendix A.

Seven cycles of interactive validation and 200 response ordering tests were used to examine the model quality and prevent the OPLS Da model from overfitting. The response ordering test plots for the validity of the OPLS Da model are shown in Appendix A. The R2 and Q2 values of the OPLS Da model were linearly regressed on the R2Y and Q2Y axes of the original model, and the intercept values of regression straight lines with the y-axis were labeled R2 and Q2, respectively. The results of the replacement test plot showed that the Q2 values were lower than the original points and the Q2 value was less than zero, indicating that the established OPLS Da model was reasonable.

An S-plot loading plot of the OPLS Da model was used to characterize the distribution of metabolites among the sample comparison groups and the intensity of the metabolites’ effects on the comparison groups. The metabolite distribution S-plot loading plot of the sample comparison groups is shown in Appendix A. The horizontal coordinates of the S-plot loading plot were the eigenvalues of the metabolite effects on the sample comparison groups. The vertical coordinates were the correlations between the sample scores and the metabolites. The differences in the metabolites near the upper-right and lower-left corners of the S-plot loading plot were significant. The results showed that the primary difference metabolites were consistent for groups B–H vs. CK. At the same time, the compound types were slightly different between the other sample comparison groups (groups vs. A, groups vs. B, groups vs. C, groups vs. D, groups vs. E, and groups vs. F), except for cholic acid (in the S-plot loading plot, metabolite number: 8.80_408.2878n), N-(1-deoxy-1-fructosyl) isoleucine (1.32_293.1477n), N-(1-deoxy-1-fructosyl) leucine (0.82_293.1477n), N-(1-deoxy-1-fructosyl) phenylalanine (2.14_327.1321n), and N-(1-deoxy-1-fructosyl) valine (0.78_279.1321n). Other glycosylated amino acids were the main compounds that caused the differences between the comparison groups (Appendix A).

Differentially expressed metabolites (DEMs) among sample comparison groups were screened using the following criteria of the OPLS Da model: first principal component VIP of >1 and a *t*-test *p*-value of <0.05. The distributions of DEMs among the sample comparison groups are shown in Appendix A and Appendix A. The results showed that the total number of DEMs between the sample comparison groups A–H vs. CK was 674–705, and the total number of DEMs between the two-by-two oral–gastrointestinal digestion sample comparison groups (A–H) showed an increasing trend with an increase in the difference in digestion time among the comparison sample groups.

To clarify the metabolic markers of the digestion products and their metabolic pathways, the analysis was focused on the DEMs between the digested and undigested sample groups (groups vs. CK). Sorted according to the variable weight values of VIP in the sample comparison groups, information on the top 20 DEMs and their correlations between the groups and the CK sample comparison groups is shown in Figure 9 and Appendix A. The results showed that the correlation between the top 20 DEMs and the eight sample comparison groups was higher than 0.95. There were 16 DEMs with consistent information, and there was a difference in the contribution of DEMs to the sample comparison groups. The common DEMs were cholic acid analogs and glycosylated amino acid compounds. Their weight values in the sample comparison groups ranged from 6 to 65 Da, among which cholic acid, THA, 3alpha-hydroxy-5beta-chola-7,9(11)-dien-24-oic acid, N-(1-deoxy-1-fructosyl) isoleucine, murocholic acid, N-(1-deoxy-1-fructosyl) phenylalanine, N-(1-deoxy-1-fructosyl) leucine, ursocholic acid, and PI (16:0/18:0) had weight values of VIP greater than 10 for the groups vs. CK. The above nine DEMs were the key DEMs in the digestion products. The distribution of the nine DEMs among the sample comparison groups is shown in Appendix A. N-(1-deoxy-1-fructosyl) isoleucine, N-(1-deoxy-1-fructosyl) phenylalanine, and N-(1-deoxy-1-fructosyl) leucine were highly expressed in the undigested samples and minimally expressed in the digestion products, suggesting that these three compounds were degraded in the oral–gastrointestinal simulated digestion of the protein–peptide-based materials. The other six DEMs were highly expressed in the digestion products, suggesting they were new DEMs generated after the digestion of the base materials.

Although the nine VIP > 10 compounds described above were metabolites that were significantly differentially expressed between the sample comparison groups, the KEGG pathways they were enriched in were not significantly enriched for most DEMs. Therefore, pathway enrichment analysis was performed for all VIP > 1 DEMs, and the results of the KEGG pathways analysis showed significantly enriched sample comparison groups, which are shown in Figure 10 and Appendix A. The results showed that valine, leucine, and isoleucine biosynthesis, arginine and proline metabolism, beta-alanine metabolism, starch and sucrose metabolism, sphingolipid metabolism, pantothenate and CoA biosynthesis, aminoacyl-tRNA biosynthesis, and ABC transporters were upregulated pathways that were significantly enriched for DEMs in the comparison group of digestion product samples (*p*-value < 0.05). The downregulated pathways included the citrate cycle (TCA cycle), alanine, aspartate, and glutamate metabolism, arginine and proline metabolism, histidine metabolism, glyoxylate and dicarboxylate metabolism, and aminoacyl-tRNA biosynthesis, which were significantly enriched for DEMs in the comparison group of digestion product samples (*p*-value < 0.05). Notably, the arginine and proline metabolism pathways and the aminoacyl-tRNA biosynthesis pathway were upregulated and downregulated for multiple DEMs. Between the comparison groups of digestion products with predominantly intestinal digestion (A–C vs. CK), the arginine and proline metabolism pathway showed more significance as a downregulated pathway than as an upregulated pathway (downregulated *p*-value < upregulated *p*-value). In contrast, the carbamoyl-tRNA biosynthesis pathway showed the opposite trend, and its upregulated pathway was highly significant (*p*-value < 0.01). Between the comparison groups of digestion products with predominantly gastric digestion (F–H vs. CK), both pathways were more upregulated than downregulated, and the upregulated pathways were highly significant (*p*-value < 0.01). It was hypothesized that focusing on the upregulated pathways of DEM enrichment in digestion products during the pre-digestion period and the downregulated pathways of DEM enrichment during the post-digestion period could provide insights into the oral–gastrointestinal digestion of the protein–peptide-based materials, as well as the synthesis and metabolism of nutritional functional components.

The number distribution of DEMs annotated to the significantly enriched KEGG pathways is shown in Appendix A. The results showed that the number of DEMs annotated on the significantly enriched KEGG pathway was between two and five. In the digestion products of the A–C vs. the CK comparison groups, the upregulated pathways of aminoacyl-tRNA, ABC transporter, pantothenic acid, and coenzyme A biosynthesis, and the downregulated pathways of arginine and proline metabolism were KEGG-enriched pathways annotated with more DEMs. The number of compounds annotated was four or five. In the digestion products of the F–H vs. CK comparison groups, in addition to the above upregulated pathways, arginine and proline metabolism, as an upregulated pathway, was also annotated to four DEMs, and the types of DEMs annotated as downregulated pathways were consistent with those of the A–C vs. CK comparison groups.

The DEMs annotated in the significantly enriched KEGG pathways and their correlations were parsed to mine DEM markers. The information regarding the DEMs between the sample comparison groups is shown in Appendix A. The results showed that 17 metabolites, including L-glutamic acid, L-lysine, L-arginine, L-leucine, citric acid, isocitrate, sphingosine, L-isoleucine, spermine, trehalose, imidazole acetic acid ribotide, L-histidine trimethyl betaine, agmatine, 4-guanidinobutanoic acid, pyrroline hydroxycarboxylic acid, subaphylline, and L-phenylalanine, were the key DEMs in the enriched KEGG pathway. Their contributed weight values ranged from 1 to 7.8 for the sample comparison group. Considering that the amino acid anabolic pathways were the main enrichment pathways in the oral–gastrointestinal digestion of the base materials and that the essential amino acids L-isoleucine and L-leucine annotated in the biosynthesis pathway of valine, leucine, and isoleucine (https://www.genome.jp/entry/map00290, accessed on 16 February 2024) and aminoacyl-tRNA biosynthesis pathway (https://www.genome.jp/entry/map00970, accessed on 16 February 2024) were metabolites with significant contributions (VIP > 4) to the sample comparison group of the digestion products, we focused on the correlations between L-isoleucine and L-leucine and other metabolites. The correlations between the DEMs are shown in Appendix A. The results showed that in the digestion products of A–C vs. CK comparison groups, the components of trehalose, L-lysine, L-arginine, L-histidine, riboflavin, sphingosine, spermine, sphinganine, phytosphingosine, L-valine, and 4-guanidino butanoic acid, with L-isoleucine and L-leucine varied consistently, with a positive correlation between the metabolites. In contrast, L-glutamic acid and L-phenylalanine were negatively correlated with L-isoleucine and L-leucine. In the digestion products of E–H vs. CK comparison groups, L-isoleucine was negatively correlated with L-leucine, and L-leucine was positively correlated with L-glutamic acid, L-phenylalanine, trehalose 6-phosphate, subaphylline, and niacinamide. In contrast, L-isoleucine remained positively correlated with L-lysine and L-arginine, which was consistent with the information on positively correlated metabolites in the digestion products of A–C vs. CK comparison groups. The main reason for the shift in the correlation between L-isoleucine and L-leucine in oral–gastrointestinal digestion was that L-isoleucine was always metabolized in oral–gastrointestinal digestion products. In contrast, L-leucine was more frequently metabolized than synthesized in the gastric digestion products and more often synthesized than metabolized in the intestinal digestion products. The DEMs positively correlated with L-isoleucine, and L-leucine might be the main component involved in the subsequent nutrient supply to the organism.

In summary, nine types of bile acid analogs and glycosylated amino acid metabolites (VIP > 10) and 17 metabolites, including L-isoleucine and L-leucine (1 < VIP < 7.8), can be used as DEM markers for monitoring protein–peptide-based material digestion, and 14 pathways, including arginine and proline metabolism and aminoacyl-tRNA biosynthesis, can be used for monitoring base material digestion for the production pathways of DEM markers.

## 4. Discussion

Usually, some differences exist between the *in vitro* ACE inhibitory activity and the *in vivo* antihypertensive activity of protein–peptide-based materials. Protein–peptide-based materials with better ACE inhibitory activity may exhibit reduced antihypertensive activity after oral–gastrointestinal digestion *in vivo*. Conversely, protein–peptide-based materials with relatively poor ACE inhibitory activity show better antihypertensive activity after oral–gastrointestinal digestion *in vivo*, possibly due to the fact that protein–peptides are degraded during the digestion process to produce new peptide molecules with higher ACE inhibitory activity. Tian [38] investigated the effects of commercial protease hydrolysis and *in vitro* gastrointestinal digestion simulation on the release of ACE inhibitory peptides from defatted goat milk powder. Neutrase, alcalase, protamex, and flavourzyme protease were used for the enzymatic hydrolysis of defatted goat milk powder, and the hydrolyzed products were subjected to gastrointestinal digestion simulation *in vitro*. Then, the peptide content, ACE inhibitory activity, and molecular weight distribution of the gastrointestinal digestion products were analyzed. It was found that the peptide content and ACE inhibitory activity of the gastrointestinal digestion products decreased after enzymatic hydrolysis with alcalase and flavourzyme protease, whereas the peptide content and ACE inhibitory activity of the digestion products increased slightly after enzymatic hydrolysis with protamex protease, and the peptide content and ACE inhibitory activity of the digestion products increased significantly after enzymatic hydrolysis with neutrase protease. The molecular weight distribution showed that the proportion of peptides of <1000 Da increased significantly after gastrointestinal digestion *in vitro*. Yang [35] found that rice hydrolysate products obtained by enzymatic hydrolysis with neutrase protease (8% protein hydrolysis degree) had significantly higher ACE inhibitory activities and increased contents of 200–500 Da peptides after gastrointestinal digestion. Cui [36] found that when milk protein was hydrolyzed by neutral protease at a hydrolysis degree of 17%, the peptide content and ACE inhibitory activity of the products were significantly improved after 4 h of gastrointestinal digestion of the milk hydrolysis products, and the percentage of 200–500 Da peptides in the digestion products was significantly increased. In the present study, after oral–gastrointestinal digestion of the protein–peptide-based materials, the percentage of products with molecular weights of <180 Da increased, and that of products with molecular weights of 180–500 Da decreased compared with those in the undigested control samples; these findings were consistent with the results of the gastrointestinal digestion of goat’s milk powder hydrolyzed by alkaline protease. Moreover, these results were presumably due to the hydrolysis of the protein–peptide-based materials by alkaline protease, which has the same hydrolysis site as pepsin. Both alkaline and pepsin proteases have hydrophobic amino acid cleavage sites, which result in fractions with molecular weights ranging from 180 to 500 Da produced by alkaline protease. These products were further degraded by pepsin protease to produce fractions of <180 Da.

## 5. Conclusions

*S. rugosoannulata* protein–peptide-based materials produced abundant peptide molecules with good ACE inhibitory activity and an IC_50_ value of 0.004–0.096 mg/mL after oral–gastrointestinal digestion. Lipid and lipid-like molecules; fatty and carboxylic acids and their derivatives; and amino acids, peptides, and their analogs were the major digestion metabolite taxa, and bile acids and glycosylated amino acids were the main differential compounds between the digestion sample groups. DEMs were significantly enriched in the amino acid synthesis and energy metabolism pathways, such as valine, leucine, and isoleucine biosynthesis; arginine and proline metabolism; aminoacyl-tRNA biosynthesis; and citric acid cycle (tricarboxylic acid cycle). These were the hallmark pathways for the production of DEMs via the digestion of protein–peptide-based materials. The glycosylated amino acids, bile acids, and essential amino acids L-isoleucine and L-leucine were critical DEMs in the *S. rugosoannulata* mushroom protein–peptide digestion product.

## Figures and Tables

**Figure 1 foods-13-02546-f001:**
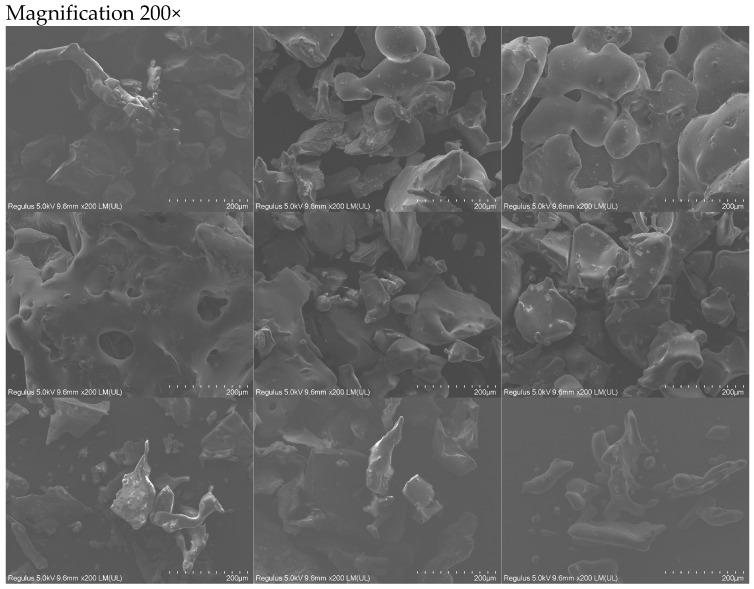
SEM plots of base material digestion products. Note: Magnification from top to bottom: 200×, 500×, 1000×, 2000×. Digestion products from left to right: CK (undigested products), digestion products 1 to 8 (described in Section 2.2).

**Figure 2 foods-13-02546-f002:**
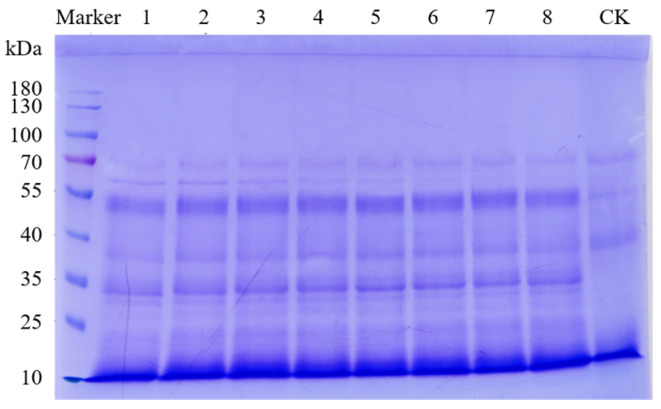
SDS-PAGE plot of base material digestion products. Note: SDS-PAGE lane information represents marker, digestion products 1 to 8, and undigested control sample (CK).

**Figure 3 foods-13-02546-f003:**
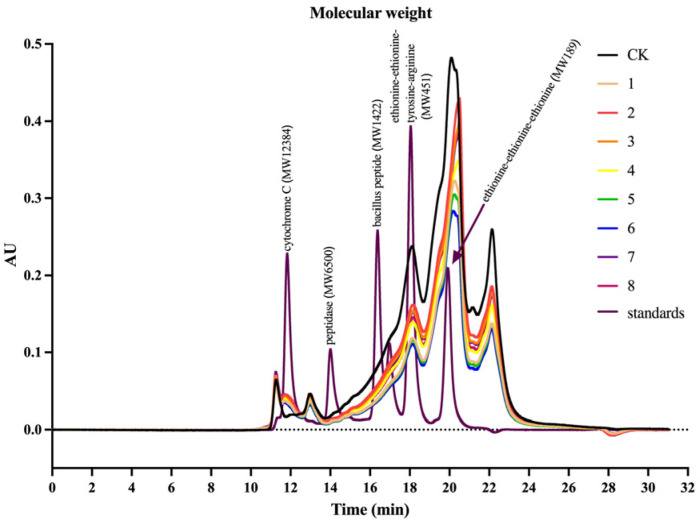
Molecular weight distribution plot of base material digestion products. Note: The legend information represents digestion products 1 to 8, undigested control sample (CK), and molecular weight-corrected standards.

**Figure 4 foods-13-02546-f004:**
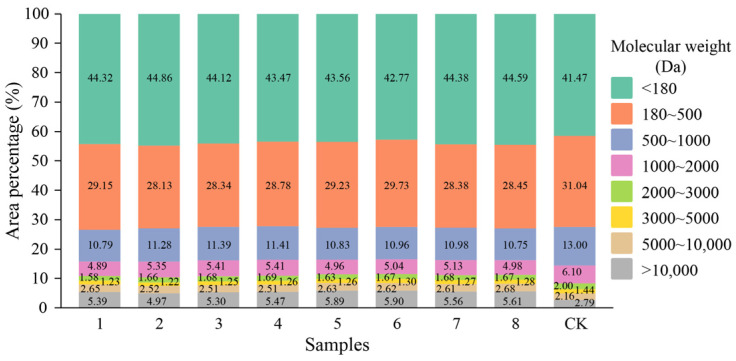
Molecular area percentage of base material digestion products. Note: The horizontal coordinate sample information represents digestion products 1 to 8 and undigested control sample (CK).

**Figure 5 foods-13-02546-f005:**
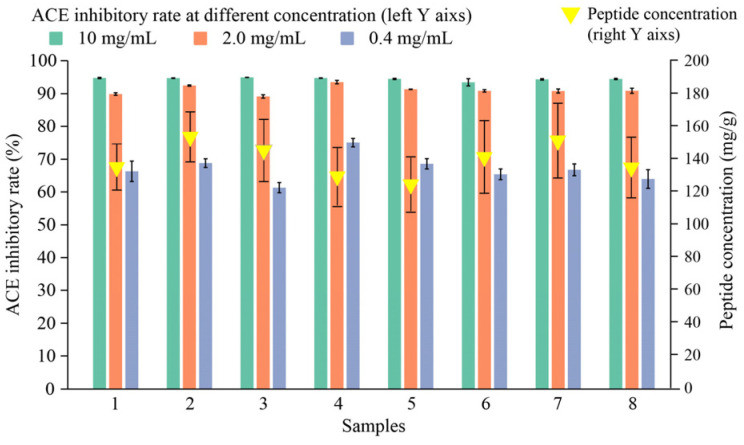
Peptide concentration and ACE inhibition rates of base material digestion products. Note: The horizontal coordinate sample information represents digestion products 1 to 8.

**Figure 6 foods-13-02546-f006:**
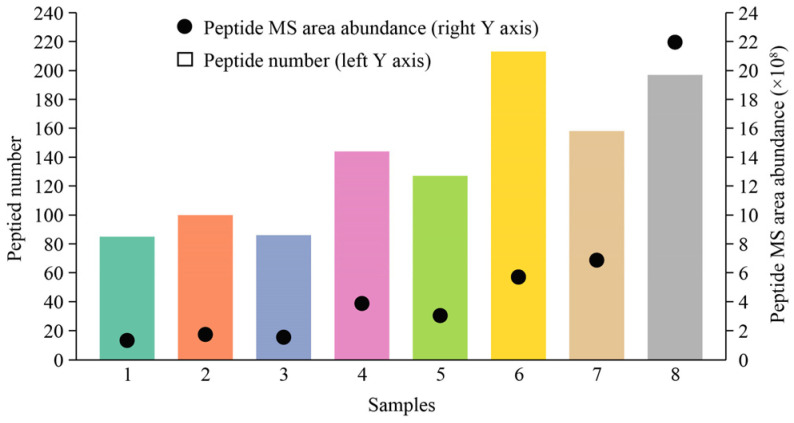
Peptide spectral distribution of base material digestion products. Note: The horizontal coordinate sample information represents digestion products 1 to 8.

**Figure 7 foods-13-02546-f007:**
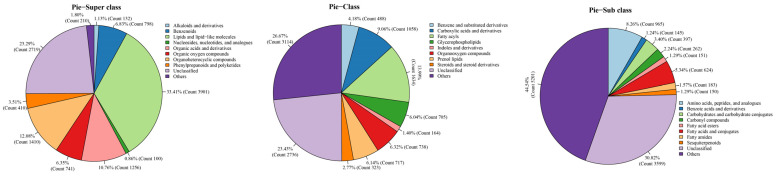
Classification of metabolites.

**Figure 8 foods-13-02546-f008:**
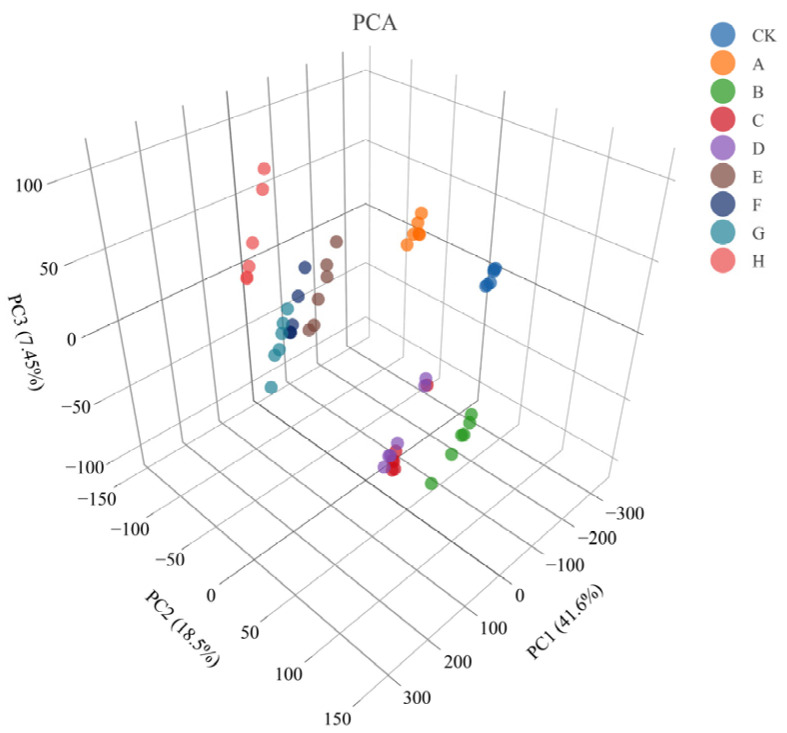
PCA of digestion products. Note: Legend information represents digestion products A to H and undigested control sample (CK) described in Section 2.2.

**Figure 9 foods-13-02546-f009:**
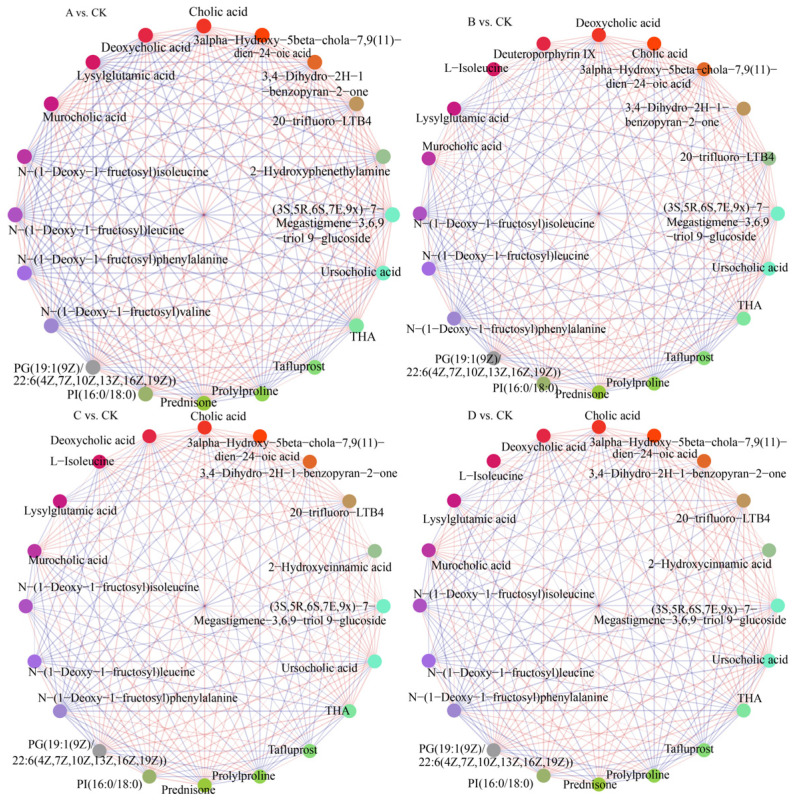
Correlation network of top 20 DEMs between groups and CK sample comparison groups.

**Figure 10 foods-13-02546-f010:**
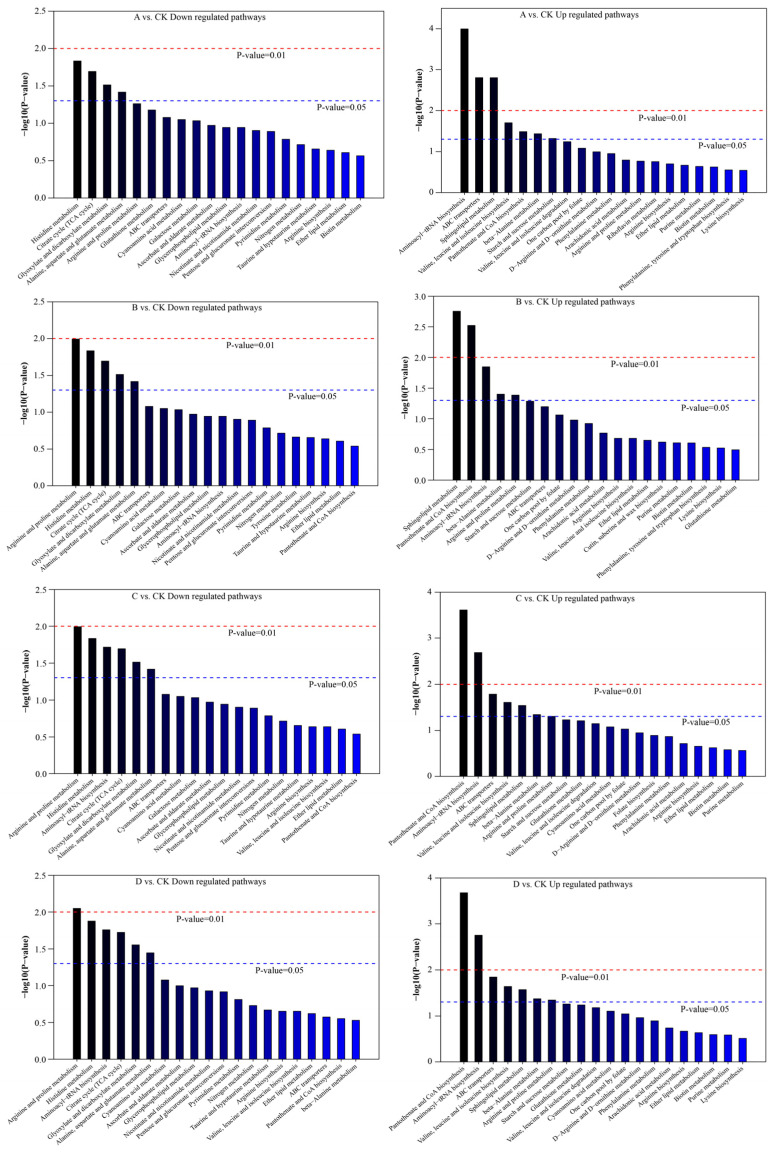
KEGG enrichment pathways between sample comparison groups.

## Data Availability

The original contributions presented in the study are included in the article/Appendix A, further inquiries can be directed to the corresponding author.

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
