# Peer review of "Combined Peptidomics and Metabolomics Analyses to Characterize the Digestion Properties and Activity of Stropharia rugosoannulata Protein–Peptide-Based Materials"

_foods, 2024, doi:10.3390/foods13162546_

Round 1

Reviewer 1 Report

Comments and Suggestions for Authors

Here are some potential issues in the Introduction that could be addressed for clarity and coherence:

1.     There is redundancy in stating the benefits and activities of protein-peptides multiple times (e.g., antioxidant, anti-tumor, anti-inflammatory). Consider consolidating these into a concise list or grouping related activities together.

2.     The introduction lacks a clear structure or flow. It jumps between describing the general properties of protein-peptides, specific findings from previous studies, and the objectives of the current study without a smooth transition.

3.     Some technical terms and abbreviations (e.g., ACE inhibition, IC50) are introduced without prior explanation. Ensure that such terms are defined or explained upon first use for readers unfamiliar with the field.

4.     The introduction covers a broad range of potential applications and benefits of protein-peptides without clearly narrowing down to the specific focus of the current study until later paragraphs. Consider starting with a more focused statement about the specific aims of the study.

5.     Improve the flow between sentences and paragraphs to ensure a coherent narrative. Each paragraph should logically lead into the next, building a cohesive argument for the importance of the study.

6.     The specific objectives of the current study (analyzing digestion properties, metabolic pathways, etc.) are introduced late in the introduction. Consider moving these objectives earlier to provide clearer context for the study.

7.     Some sentences are long and complex, making it challenging to follow the main points. Break down longer sentences into smaller, more digestible units to improve readability.

8.     Some details, such as specific findings from previous studies, could be summarized more briefly to focus more on the current study's rationale and objectives.

9.     The introduction could benefit from a concluding paragraph that summarizes the gaps in knowledge and transitions smoothly into the specific aims and methods of the current study.

Addressing these issues will help to improve the clarity, organization, and impact of the Introduction section, ensuring that it effectively introduces the study's background, rationale, and objectives to the reader.

Here are some issues that could be addressed in the "Materials and Methods" section to improve clarity and detail:

1.     The description of how the S. rugosoannulata protein-peptide base materials were prepared is somewhat fragmented and lacks a clear step-by-step explanation. It would be helpful to clearly outline each step (e.g., mushroom concentration, enzyme addition, ultrasonic conditions, filtration process) in a sequential manner.

2.     While the overall procedure for in vitro simulation digestion is described, specific details such as the exact composition of each digestive solution (saliva-simulating, gastric-digestion, intestinal-digestion) are mentioned briefly. Providing more specific details about the composition or reference to a standard protocol used for these solutions would enhance reproducibility.

3.     Ensure consistency in naming conventions and abbreviations throughout the section. For instance, "simulated" vs. "simulating" digestion is used interchangeably; standardizing this terminology can avoid confusion.

4.     When discussing the molecular weight distribution analysis (SDS-PAGE and HPLC), ensure that details such as the concentrations of digestion products used, standards employed, and specific instrument settings are clearly specified. This helps in replicating the experiments accurately.

5.     For ACE inhibitory activity analysis and peptidomics/metabolomics, describe the analytical methods in more detail. Include specifics on how data were collected, processed, and interpreted (e.g., software used, parameters for analysis).

6.     Some paragraphs are quite dense and technical. Breaking down complex procedures into smaller, more digestible steps and using bullet points or numbered lists can improve readability and comprehension.

7.     Ensure that references to previous literature (e.g., [30], [28]) are consistently linked to specific procedures or methods they pertain to. This helps in validating methodologies and findings.

8.     Expand on how QC samples were prepared and utilized throughout the experiment to ensure reliability and reproducibility of results.

9.     Improve the flow between different subsections (e.g., from digestion simulations to morphological analysis) to ensure a logical progression of methods from one step to the next.

10.  By addressing these issues, the "Materials and Methods" section can be enhanced to provide a clearer, more detailed account of the experimental procedures, ensuring reproducibility and transparency in scientific research.

Comments on the Quality of English Language

Moderate editing of english is required

Author Response

Thanks for the reviewers' comments concerning our manuscript (Manuscript ID: foods-3125848). We have studied the comments carefully and made corrections, which we hope to meet with approval. We ask Prof. Samantha to check the language of the manuscript. Revised portions are marked in red on the paper. The main corrections in the paper and the responses to the reviewer's comments are as follows:

Reviewer 1: Comments and Suggestions for Authors

1.     There is redundancy in stating the benefits and activities of protein-peptides multiple times (e.g., antioxidant, anti-tumor, anti-inflammatory). Consider consolidating these into a concise list or grouping related activities together.

Thanks for the reviewer's comments. The following table summarizes the literature cited in the article on peptide activity. We have supplemented the table below as Supplementary Table 1 of the manuscript.

2.     The introduction lacks a clear structure or flow. It jumps between describing the general properties of protein-peptides, specific findings from previous studies, and the objectives of the current study without a smooth transition.

Thanks for the reviewer's comments. It is revised. Please check the revised Introduction section for details.

3.     Some technical terms and abbreviations (e.g., ACE inhibition, IC50) are introduced without prior explanation. Ensure that such terms are defined or explained upon first use for readers unfamiliar with the field.

Thanks for the reviewer's comments. It is revised. Please check the revised Introduction section for details.

4.     The introduction covers a broad range of potential applications and benefits of protein-peptides without clearly narrowing down to the specific focus of the current study until later paragraphs. Consider starting with a more focused statement about the specific aims of the study.

Thanks for the reviewer's comments. It is revised. Please check the revised Introduction section for details.

5.     Improve the flow between sentences and paragraphs to ensure a coherent narrative. Each paragraph should logically lead into the next, building a cohesive argument for the importance of the study.

Thanks for the reviewer's comments. It is revised. Please check the revised Introduction section for details.

6.     The specific objectives of the current study (analyzing digestion properties, metabolic pathways, etc.) are introduced late in the introduction. Consider moving these objectives earlier to provide clearer context for the study.

Thanks for the reviewer's comments. It is revised. Please check the revised Introduction section for details.

7.     Some sentences are long and complex, making it challenging to follow the main points. Break down longer sentences into smaller, more digestible units to improve readability.

Thanks for the reviewer's comments. It is revised. Please check the revised manuscript for details.

8.     Some details, such as specific findings from previous studies, could be summarized more briefly to focus more on the current study's rationale and objectives.

Thanks for the reviewer's comments. It is revised. Please check the revised Introduction section for details.

9.     The introduction could benefit from a concluding paragraph that summarizes the gaps in knowledge and transitions smoothly into the specific aims and methods of the current study.

Thanks for the reviewer's comments. It is revised. Please check the revised Introduction section for details. 

Here are some issues that could be addressed in the "Materials and Methods" section to improve clarity and detail:

1.     The description of how the S. rugosoannulata protein-peptide base materials were prepared is somewhat fragmented and lacks a clear step-by-step explanation. It would be helpful to clearly outline each step (e.g., mushroom concentration, enzyme addition, ultrasonic conditions, filtration process) in a sequential manner.

Thanks for the reviewer's comments. The relevant information is described in 2.1, and we have re-added the mushroom and protease information as detailed in the revised manuscript.

2.     While the overall procedure for in vitro simulation digestion is described, specific details such as the exact composition of each digestive solution (saliva-simulating, gastric-digestion, intestinal-digestion) are mentioned briefly. Providing more specific details about the composition or reference to a standard protocol used for these solutions would enhance reproducibility.

Thanks for the reviewer's comments. We have made relevant additions, as detailed in the revised manuscript and Supplementary Table 2.

3.     Ensure consistency in naming conventions and abbreviations throughout the section. For instance, "simulated" vs. "simulating" digestion is used interchangeably; standardizing this terminology can avoid confusion.

Thanks for the reviewer's comments. It is revised. Please check the revised manuscript.

4.     When discussing the molecular weight distribution analysis (SDS-PAGE and HPLC), ensure that details such as the concentrations of digestion products used, standards employed, and specific instrument settings are clearly specified. This helps in replicating the experiments accurately.

Thanks for the reviewer's comments. Relevant information is described in 2.4, including information on the analytical instrument and standards used (standard information is also described in 2.1). Calculations show that the up-sampling concentration was 50 mg/mL for SDS-PAGE and 10 mg/mL for HPLC. 

5.     For ACE inhibitory activity analysis and peptidomics/metabolomics, describe the analytical methods in more detail. Include specifics on how data were collected, processed, and interpreted (e.g., software used, parameters for analysis).

Thanks for the reviewer's comments. We have made relevant additions, as detailed in the revised manuscript.

6.     Some paragraphs are quite dense and technical. Breaking down complex procedures into smaller, more digestible steps and using bullet points or numbered lists can improve readability and comprehension.

Thanks for the reviewer's comments. Please check the revised manuscript.

7.     Ensure that references to previous literature (e.g., [30], [28]) are consistently linked to specific procedures or methods they pertain to. This helps in validating methodologies and findings.

Thanks for the reviewer's comments. We have confirmed the above information.

8.     Expand on how QC samples were prepared and utilized throughout the experiment to ensure reliability and reproducibility of results.

Thanks for the reviewer's comments. QC sample is a mixture of all samples in equal parts after pre-processing, usually 10 µL or 20 µL of each sample. During the mass spectrometry onboarding process of samples, QC samples will be interspersed between samples, equivalent to making a technical duplicate, used to evaluate the stability of the mass spectrometry platform throughout the onboarding process.

9.     Improve the flow between different subsections (e.g., from digestion simulations to morphological analysis) to ensure a logical progression of methods from one step to the next.

Thanks for the reviewer's comments. Morphology is the most intuitive analysis of lyophilized digestion products, and this section is also relatively simple, so we placed it at the top of the analysis.

10.  By addressing these issues, the "Materials and Methods" section can be enhanced to provide a clearer, more detailed account of the experimental procedures, ensuring reproducibility and transparency in scientific research.

Thanks for the reviewer's comments. The Materials and Methods section has been modified to better replicate the experiment.

Reviewer 2 Report

Comments and Suggestions for Authors

This manuscript presents some data on the in vitro digestion of some previously prepared and characterised mushroom base material.

Because of poor english, no detail given on digestion protocols used, poor resolution on some figures, poor statistical treatment on the data presented and poor discussion and conclusion, I am of the opinion that this manuscript cannot be published without extensive revision.

I include below some non exhaustive questions that would need answering in this revision process 

Please define the term "protein-peptide base material" as this is not a commonly used terminology.

Line 56 Benadryl is a registered trademark. Use the generic term Diphenhydramine instead, which is an antihistamine and not a blood pressure lowering drug (actually, antihistamines are known to ellicit increases in blood pressure) 

Lines 94-97 Please define the simulated digestive fluids, including pH, slats, and enzymes concentrations. At present, it is impossible for the reader to evaluate the conditions used and compare them to exisiting literature protocols

Lines 109-120 Please indicate how the digestions steps were stopped for the 1-8 test groups. Did you change pH, use protease blockers etc?

Section 3.1 SEM morphology. Please indicate the meaning of this experiment. This result does not seem to add anything to the discussion, consider removing this section, together with the associated figures.

On figure 4, it is not clear how the athors determined the various moelcular weights categories from the data presented on figure 3. What model time-Mw was used? and was there statistical analysis done to evaluate differences between samples? The control sample seem to contain less intact protein (2.79%) and same amount of small peptides (72.5% <500Da) as digested samples

Section 3.3 Please discuss the assumption that ACE inhibition is solely due to petides in the samples. This seems contradictory to results obtained

Section 3.4 No statistical analysis is presented on figure 6, which makes the interpretation of results (and talk of "fluctuated wildly" line 257) impossible

Section 3.5 Figure 7, 9 and 10 are very poor resolution and impossible to read. Figure 8 has a different and unexplained legend. This makes understanding of this section near impossible. Please revise. Also, as 33% of metabolites identified are lipids, please indicate in the introduction the lipids content in the undigested samples and indicate if lipases were used in the digestion protocols used. Authors talk of cholic acids derivatives without proposing an explanation for their presence. Was a negative control done on the digestive fluids used, as these could easily contain bile acids? Also, authors talk of glycosilated amino acids, without offering a discussion as to their origin in the undigested sample.

Section 3.7 Discussion centred on non-digestive enzyme hydrolysis of protein material cannot be equated to digestion, please rewrite this section. Also, the authors use a protein material that was originally hydrolysed using alcalase, and no mention of that point is made in this section, please revise.

Comments on the Quality of English Language

The English language is quite poor throughout the abstract and introduction and needs editing

Author Response

Thanks for the reviewers' comments concerning our manuscript (Manuscript ID: foods-3125848). We have studied the comments carefully and made corrections, which we hope to meet with approval. We ask Prof. Samantha to check the language of the manuscript. Revised portions are marked in red on the paper. The main corrections in the paper and the responses to the reviewer's comments are as follows:

Reviewer 2 Comments and Suggestions for Authors

Please define the term "protein-peptide base material" as this is not a commonly used terminology.

Thanks for the reviewer's comments. Protein-peptide base material is a mixture with a high protein and peptide content, and we have added information to this section. The following literature has been reported on the study of protein peptides:

  1. Ma, J.W.; He, L.Y.; Cai, J.X.; Xu, L.B.; Cao, S.Q.; Qi, X.Y. Preparation and characterization of Pneumatophorus japonicus protein peptide. Food Fermentation Ind. 2022, 48, 204-212, doi:10.13995/j.cnki.11-1802/ts.029039.
  2. Yu, L.N.; Yang, Q.L.; Feng, J.X.; Sun, J.; Bi, J.; Zhang, C.S. Preparation and antioxidant activities of peanut (Arachin conarachin L.) protein peptides by lactobacillus solid state fermentation method. In Proceedings of the Mechanical Components and Control Engineering III, 2014, 1573-+, doi: 10.4028/www.scientific.net/AMM.668-669.
  3. Du, Y.L.; Yan, J.G.; Li, X.M.; Yu, Q.; Li, C.H.; Mao, X.L. Evaluate on nutritional value and efficacy of fermented soybean protein peptide powder. Food Fermentation Ind. 2022, 48, 54-58, doi:10.13995/j.cnki.11-1802/ts.029325.
  4. Wang, S.Q.; Chen, X.J.; Liang, L.P.; Yang, N.; Xuan, L.; Li, C.;Y. Li, H.G.; Quna, Z.Z.; Liu, X.J. Effects of dietary enzymatic protein peptide on growth performance, serum immune indices and intestinal flora of Weaned Piglets. Chin. J. Anim. Nutr. 2022, 34, 839-851.
  5. Gu, H.F.; Gao, J.; Shen, Q.; Gao, D.X.; Wang, Q.; Tangyu, M.Z.; Mao, X.Y. Dipeptidyl peptidase-IV inhibitory activity of millet protein peptides and the related mechanisms revealed by molecular docking. LWT-Food Sci. Technol. 2021, 138, 110587, doi:10.1016/j.lwt.2020.110587.

Line 56 Benadryl is a registered trademark. Use the generic term Diphenhydramine instead, which is an antihistamine and not a blood pressure lowering drug (actually, antihistamines are known to ellicit increases in blood pressure) 

Thanks for the reviewer's comments. We have corrected the misspelling. This part of the study was validated in animal studies and published in Food, 2023, 12, 3461; Food Funct, 2024, 15, 5527.

Lines 94-97 Please define the simulated digestive fluids, including pH, slats, and enzymes concentrations. At present, it is impossible for the reader to evaluate the conditions used and compare them to exisiting literature protocols

Thanks for the reviewer's comments. We have made relevant additions, as detailed in the revised manuscript and Supplementary Table 2.

Lines 109-120 Please indicate how the digestions steps were stopped for the 1-8 test groups. Did you change pH, use protease blockers etc?

Thanks for the reviewer's comments. We did not terminate the digestion reaction by adjusting the pH or using a protease blocker, mainly because it is also impossible to terminate the digestion reaction by adjusting the pH or using a protease blocker during real in vivo digestion. After we collected the digestion products at different digestion levels, we first snap-froze the digestion product solution in liquid nitrogen and then freeze it at -70°C for 48 h. After freeze-drying, we still stored the samples at -70°C, which inhibited the enzyme activity in the digestion products to a certain extent. When performing subsequent targeted and untargeted metabolite analyses of the digestion products, the lyophilized powder of the digestion products was also stored at -70°C to -40°C, which reduced the effect of the digestion enzymes on the products.

Section 3.1 SEM morphology. Please indicate the meaning of this experiment. This result does not seem to add anything to the discussion, consider removing this section, together with the associated figures.

Thanks for the reviewer's comments. Morphology is the most intuitive analysis of lyophilized digestion products, and it can characterize the digestive properties of the mushroom protein-peptide base materials (whether they are easily digested or not). Because this part is relatively simple, we did not include it in the discussion.

On figure 4, it is not clear how the athors determined the various moelcular weights categories from the data presented on figure 3. What model time-Mw was used? and was there statistical analysis done to evaluate differences between samples? The control sample seem to contain less intact protein (2.79%) and same amount of small peptides (72.5% <500Da) as digested samples

Thanks for the reviewer's comments.

Figure 3 shows the different molecular weight bands based on the standards' peaks; the same case is shown in the following literature.

  1. Su, G.W.; Ren, J.Y.; Zhao, M.M.; Sun, D.W. Comparison of Superdex Peptide HR 10/30 Column and TSK Gel G2000 SWXL Column for Molecular Weight Distribution Analysis of Protein Hydrolysates. Food Bioprocess Technol. 2013, 6, 3620-3626. https://doi.org/10.1007/s11947-012-0965-8.
  2. Zhong, Y.; Zhou, Y.; Ma, M.; Zhao, Y.; Xiang, X.; Shu, C.; Zheng, B. Preparation, Structural Characterization, and Stability of Low-Molecular-Weight Collagen Peptides–Calcium Chelate Derived from Tuna Bones. Foods 2023, 12, 3403. https://doi.org/10.3390/foods12183403.
  3. Han, L.; Xiao, F.Q.; Hao, G.; Bi, X.F.; Huang, M.G. lsolation, purification, and structure-activity relationship of antioxidant peptides from whey protein. China Dairy Ind. 2024, 1-14. https://link.cnki.net/urlid/23.1177.TS.20240131.1652.002.
  4. Lin, Q.L.; Huang, Y.F.; Fang, H.X.; Lin, B.Y.; Miao, S., Lin, Z.H.; Deng, K.B. Preparation and Flavor Characteristics of Maillard Reacted Peptides from Hypsizygus marmoreus. Food Res. Dev. 2024, 45, 146⁃155. https://doi.org/10.12161/j.issn.1005⁃6521.2024.03.021.

Figure 4 shows the percentage of different molecular weight segments calculated from the peak area integration of Figure 3. Combined with Fig. 3, the peak area of the CK group (black line) at the macromolecule is higher. However, the peak area of the CK group's small molecular weight segments is also very high, so the total conversion down to the macromolecule proportion is low.

We supplemented the "weight average molecular weight (Mw)" and "number average molecular weight (Mn)" of the product's different molecular weight segments, which are shown in Supplementary Table 4. From the data, it can be seen that the Mw and Mn of the >10000 Da fractions are around 16000~18000 Da. To determine the molecular weight of the peptides by HPLC, it is necessary to first obtain the standard curve by measuring the standard, and then the retention time of the measured peptides is calculated by the standard curve to obtain their molecular weight. The highest molecular weight of the standard used in HPLC, cytochrome C, is a protein composed of 105 amino acids with a molecular weight of about 12.5 kDa. In contrast, most of the proteins of Stropharia rugosoannulata searched through the UniProt database comprise 200-700 amino acids. Therefore, the molecular weight determined by HPLC does not represent the intact protein of S. rugosoannulata, and 2.79% only represents the percentage of the fractions with Mw and Mn around 16000~18000 Da.

The reviewers' comments are valuable for further research. Subsequently, we will optimize the HPLC method for protein determination.

Section 3.3 Please discuss the assumption that ACE inhibition is solely due to petides in the samples. This seems contradictory to results obtained

Thanks for the reviewer's comments.

Although some peptides were degraded during digestion, 120 mg/g dry weight of peptides was still present in the final digestion products at different digestion levels. These peptide molecules resisted to protease digestion in the in vitro oral-gastrointestinal simulated digestion assay. Moreover, after peptidomics analysis, we found that many peptides with ACE-inhibitory activity existed in the product, and a diverse range of ACE-inhibitory fragments existed in the peptides, so we hypothesized that the peptides exerted ACE-inhibitory effects. We could not determine which small molecules might have ACE inhibitory effects in untargeted metabolomics. In contrast, there are many essential amino acids in the digested products, which may play a role in the nutritional supply of the body.

Section 3.4 No statistical analysis is presented on figure 6, which makes the interpretation of results (and talk of "fluctuated wildly" line 257) impossible

Thanks for the reviewer's comments. The large fluctuations here refer to the significant differences in peptide spectral peaks in different digestion products, and we have modified the description.

Section 3.5 Figure 7, 9 and 10 are very poor resolution and impossible to read. Figure 8 has a different and unexplained legend. This makes understanding of this section near impossible. Please revise. Also, as 33% of metabolites identified are lipids, please indicate in the introduction the lipids content in the undigested samples and indicate if lipases were used in the digestion protocols used. Authors talk of cholic acids derivatives without proposing an explanation for their presence. Was a negative control done on the digestive fluids used, as these could easily contain bile acids? Also, authors talk of glycosilated amino acids, without offering a discussion as to their origin in the undigested sample.

Thanks for the reviewer's comments.

The resolution of all the figures in Figures 7, 9, and 10 is 300 ppi. The figures have been reduced in the manuscript as they must be manageable. We have attached the original figures in a zip file for reviewers to check.

The information represented by the legend of Figure 8 is described in 2.2. The digestion products numbered 1~8 for targeted peptidomics analysis, A~H for non-targeted metabolomics analysis, and CK for undigested samples.

The digestion protocol did not use lipase, and information on the digestive enzymes used was refined in 2.1. We provide raw metabolomics data for reviewer review (in the zip file, file name: Metabolomics Analysis Data Matrix). The metabolomics data, which primarily provide peak areas rather than absolute amounts of metabolites, were subjected to a series of data processing steps by multivariate statistical analysis, including peak extraction, correction, normalization, and statistical analysis. These steps aim to improve the data's reliability and accuracy and reveal sample metabolite differences. Also, mushrooms do not contain high levels of lipid substances. Since we focused on proteins and peptides and did not analyze lipids, we could not state the lipid content in undigested samples in the introduction. Many of the lipids identified by metabolomics are unsaturated fatty acids, which may be responsible for the higher percentage of lipids.

Bile salts (containing bile acids and deoxycholic acid) were present in the intestinal-digestion simulated solution at 20 mM. Bile acids are produced in human digestion, which is one reason we did not discard the cholic acid analogs after the metabolomics analysis, as they are equally present in the digestion products during human digestion. Another reason we did not discard bile acids is that we did oral experiments in spontaneously hypertensive rats simultaneously, and the bile acid analogs were also found to be differentially expressed metabolites in the serum samples of the rats. Therefore, it would not be possible to fully characterize the differentially expressed metabolites in the digestive products if this class of compounds were discarded just from the presence of bile acids in the digestive fluids.

We have yet to analyze the specific source of the glycosylated amino acids. However, these substances may be produced by combining monosaccharides and amino acids while preparing the protein-peptide base materials. The specific sources of glycosylated amino acids remain to be further analyzed.

Section 3.7 Discussion centred on non-digestive enzyme hydrolysis of protein material cannot be equated to digestion, please rewrite this section. Also, the authors use a protein material that was originally hydrolysed using alcalase, and no mention of that point is made in this section, please revise.

Thanks for the reviewer's comments.

The manuscript does not include section 3.7. The discussion section is not centered around undigested samples but around in vitro simulated digestion of products obtained by enzymatic digestion, as reported in the literature. Since the present study was also carried out on the in vitro digestion of mushrooms' enzymatic digestion products, we considered similar literature comparable and discussable.

Section 2.1 described the hydrolytic enzyme (alkaline protease) used to prepare the peptide base materials.
